# Conjugative Transfer of Acute Hepatopancreatic Necrosis Disease-Causing pVA1-Type Plasmid Is Mediated by a Novel Self-Encoded Type IV Secretion System

Dehao Wang,[a,b] Liying Wang,[a,c] Dexi Bi,[d] Jipeng Song,[a] Guohao Wang,[a] Ye Gao,[a] Kathy F. J. Tang,[a] Fanzeng Meng,[a,c] Jingmei Xie,[a,e] Fan Zhang,[a] (ID) Jie Huang,[a,f] Jianliang Li,[b] (ID) Xuan Dong[a,e]

[a]Yellow Sea Fisheries Research Institute, Chinese Academy of Fishery Sciences, Laboratory for Marine Fisheries Science and Food Production Processes, Pilot National Laboratory for Marine Science and Technology (Qingdao), Key Laboratory of Maricultural Organism Disease Control, Ministry of Agriculture and Rural Affairs, Qingdao Key Laboratory of Mariculture Epidemiology and Biosecurity, Qingdao, China

[b]Shandong Agricultural University, College of Animal Science and Veterinary Medicine, Tai'an, China

[c]Shanghai Ocean University, Shanghai, China

[d]Department of Pathology, Shanghai Tenth People's Hospital, Tongji University School of Medicine, Shanghai, China

[e]Tianjin Agricultural University, Tianjin, China

[f]Network of Aquaculture Centres in Asia-Pacific, Bangkok, Thailand

**ABSTRACT** The pathogenic pVA1-type plasmids that carry *pirAB* toxin genes are the genetic basis for *Vibrio* to cause acute hepatopancreatic necrosis disease (AHPND), a lethal shrimp disease posing an urgent threat to shrimp aquaculture. Emerging evidence also demonstrate the rapid spread of pVA1-type plasmids across *Vibrio* species. The pVA1-type plasmids have been predicted to encode a self-encoded type IV secretion system (T4SS). Here, phylogenetic analysis indicated that the T4SS is a novel member of Trb-type. We further confirmed that the T4SS was able to mediate the conjugation of pVA1-type plasmids. A *trbE* gene encoding an ATPase and a *traG* gene annotated as a type IV coupling protein (T4CP) were characterized as key components of the T4SS. Deleting either of these 2 genes abolished the conjugative transfer of a pVA1-type plasmid from AHPND-causing *Vibrio parahaemolyticus* to *Vibrio campbellii*, which was restored by complementation of the corresponding gene. Moreover, we found that bacterial density, temperature, and nutrient levels are factors that can regulate conjugation efficiency. In conclusion, we proved that the conjugation of pVA1-type plasmids across *Vibrio* spp. is mediated by a novel T4SS and regulated by environmental factors.

**IMPORTANCE** AHPND is a global shrimp bacteriosis and was listed as a notifiable disease by the World Organization for Animal Health (WOAH) in 2016, causing losses of more than USD 7 billion each year. Several *Vibrio* species such as *V. parahaemolyticus*, *V. harveyi*, *V. campbellii*, and *V. owensii* harboring the virulence plasmid (designated as the pVA1-type plasmid) can cause AHPND. The increasing number of *Vibrio* species makes prevention and control more difficult, threatening the sustainable development of the aquaculture industry. In this study, we found that the horizontal transfer of pVA1-type plasmid is mediated by a novel type IV secretion system (T4SS). Our study explained the formation mechanism of pathogen diversity in AHPND. Moreover, bacterial density, temperature, and nutrient levels can regulate horizontal efficiency. We explore new ideas for controlling the spread of virulence plasmid and form the basis of management strategies leading to the prevention and control of AHPND.

**KEYWORDS** acute hepatopancreatic necrosis disease (AHPND), type IV secretion system (T4SS), conjugative transfer, pVA1-type plasmid

**Ad Hoc Peer Reviewer** (ID) Songzhe Fu, Dalian Ocean University

Address correspondence to Xuan Dong, dongxuan@ysfri.ac.cn, or Jianliang Li, lijianliangsdau@163.com.

The authors declare no conflict of interest.

*[This article was published on 19 September 2022 with errors in the affiliations. The affiliations were updated in the current version, posted on 6 October 2022.]*

**A**cute hepatopancreatic necrosis disease (AHPND), also known as early mortality syndrome (EMS), is a severe bacterial disease with cumulative mortalities of 70–100% (1–5). It results in losses of more than USD 7 billion each year (6). AHPND was listed as a notifiable disease by the World Organization for Animal Health (WOAH) in 2016 (7). During the emergence of AHPND, several strains of *Vibrio parahaemolyticus* were identified as causative pathogens (1, 5, 8), each of which contains a ∼70kb pVA1-type plasmid carrying *pirAB* toxin genes. The *pirAB* genes encode homologs of the *Photorhabdus* insect-related (Pir) toxin proteins PirA and PirB (7). Recent studies have shown that AHPND could also be caused by the infection with strains of other *Vibrio* spp., including *V. campbellii*, *V. owensii*, *V. punensis*, and *V. harveyi* (9–15). All the reported AHPND-causing *Vibrio* strains harbor the pVA1-type plasmids and are collectively abbreviated as $V_{AHPND}$.

Previous studies have observed the horizontal transfer of a pVA1-type plasmid from AHPND-causing *V. campbellii* to non-AHPND-causing *V. owensii* (16). Furthermore, our team has reported that the conjugative transfer of pVA1-type plasmids causes the formation of novel AHPND-causing *Vibrio* (17). However, the key components and transfer mechanisms mediating conjugation are still unclear. We have compared 23 reported pVA1-type plasmids and found that they all harbor 2 gene clusters that may constitute a type IV secretion system (T4SS) (17). T4SSs are versatile assemblages that promote effector translocation and/or genetic exchange with consequent impacts on pathogenesis and genome plasticity (18). The conjugative transfer mediated by T4SSs is an important mechanism for horizontal gene transfer (19). A typical conjugative transfer system in Gram-negative bacteria consists of 3 components, including (i) transferosome: the elaboration of a pilus in the donor cell that is assembled by a T4SS; (ii) relaxosome: a nucleoprotein complex formed by the relaxase and additional proteins at the origin of transfer (*oriT*) and directed to the *nic* site; (iii) type IV coupling protein (T4CP): an essential component of the T4SS connecting the transferosome and relaxosome (20–22).

In this study, we demonstrated that the T4SS in pVA1-type plasmids is a novel Trb-type member. We identified its essential components, including a *traG* as a T4CP gene and a *trbE* gene as an ATPase gene. The construction of deletion and complementation mutants of the *trbE* and *traG* proved that the T4SS mediates the conjugative transfer of pVA1-type plasmids. Moreover, we showed the T4SS could be regulated by environmental factors such as bacterial density, temperature, and nutrient levels.

## RESULTS

**The T4SS in pVA1-type plasmids is a novel Trb-type member.** We have previously reported that the published pVA1-type plasmids contained gene clusters relevant to conjugative transfer (17). In this study, analysis with Type IV Secretion System Resource (SecReT4) indicated that the composition and organization of the conjugative transfer genes were most similar to that of the Trb-type T4SSs. In the pVPGX1 plasmid, there were 14 genes annotated to be components constituting a T4SS, including *traF*, *traG*, and a *trb* cluster of 12 genes ($trbB_1$, $-B_2$, -C, -D, -E, -F, -G, -H, -I, -J, -L, and -N) (Fig. 1A and Table 1). Those genes were remarkably conserved across pVA1-type plasmids, showing 89.54%–100.00% and 71.79%–100.00% identities in nucleotide and amino acid levels, respectively (Fig. 1B and Fig. S1).

Function prediction was further conducted via SecReT4 and the protein families database (Pfam) analysis against T4SS components with known functions, followed by manual curation. (Fig. 1C and Table 1). The components $TrbB_1$, $TrbB_2$, and TrbE of T4SS were annotated as ATPases (23). Notably, the TrbE was a homolog of VirB4 that has been recognized as the most conserved T4SS component (24). The TrbD might be an inner membrane protein and mediate the export of DNA or other Trb proteins (25, 26). The TrbG, TrbH, TrbJ, and TrbL proteins were predicted as core channel components. The TrbG might form the outer membrane pore, while the TrbH might adhere to the outer membrane and form the outer membrane channel together with TrbG (27–29). The TrbL protein was probably a central component of the inner membrane channel (30, 31). The TrbC and TrbF proteins were the components of pilus found in T4SS (32,

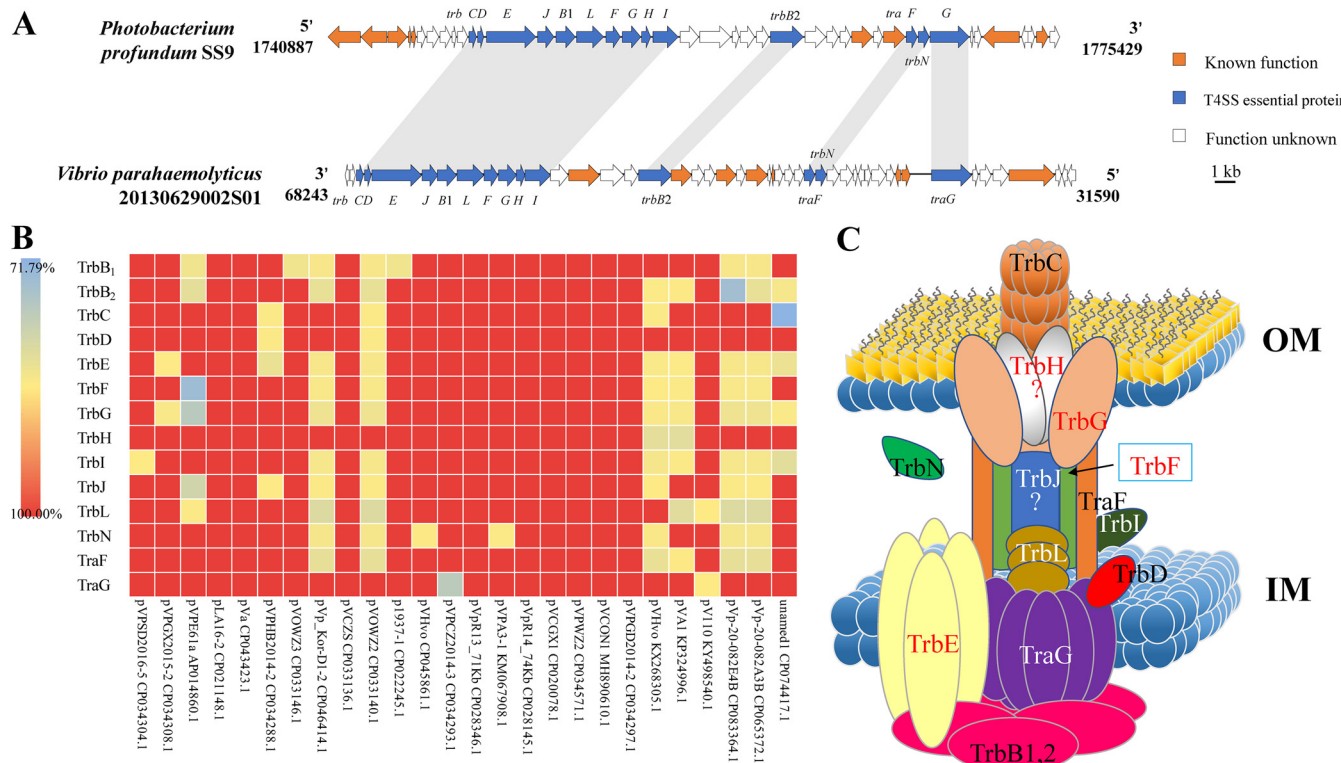

**FIG 1** Analysis and annotation of the novel Trb-type T4SS found in pVA1-type plasmids. (A) The annotation diagrams of the type IV secretion system in the pVPGX1 of *Vibrio parahaemolyticus* 20130629002S01 compared to those in chromosome 2 of *Photobacterium profundum* SS9. (B) Amino acid identities of the T4SS components across pVA1-type plasmids against the reference pVPGX1. (C) Predicted pattern of the pVA1-type plasmid-borne T4SS based on that of the VirB-type T4SS. "?" means those components have no counterparts in the VirB-type T4SS.

33). The TraF protein was the conjugative transfer signal peptidase that might process the pilin subunit of the conjugative pilus into a circular form (34). The TraG protein was the T4CP that might link the conjugative transfer DNA-relaxase complex (20, 21). The TrbI protein was not essential for conjugation but might regulate the conjugational efficiency (25). The TrbN protein was annotated as a lytic transglycosylase (35).

Phylogenetic analysis showed that T4SS of pVA1-type plasmids formed an independent branch and were closer to the *Photobacterium profundum* SS9 (Fig. 2). Indeed, the highest identity between pVPGX1 T4SS and *P. profundum* SS9 at the nucleotide level was 83.67%. The identities of the two T4SS components at the amino acid level are 30.00%–63.24% (Table 1). The above results indicated that the T4SS in pVA1-type plasmids is a novel member of Trb-type T4SS.

**Construction of the deletion and complementation mutants of *trbE* and *traG*.** To test the function of the T4SS, 2 key essential component genes, i.e., *trbE* and *traG*, were separately deleted to inactivate the T4SS in Vp2S01, resulting in deletion mutants Vp2S01Δ*trbE* and Vp2S01Δ*traG*, respectively. Deletions of these 2 genes were successfully confirmed using PCR and Sanger sequencing. Importantly, the adjacent genes remained intact and could be transcribed unaffectedly, as tested by both PCR and RT-PCR (Fig. S2). Complementation mutants Vp2S01Δ*trbE*::pRK415-*trbE* and Vp2S01Δ*traG*::pRK415-*traG* were subsequently constructed. The transcription level of the relevant genes was confirmed, which also exhibited long-term stability as the expression was well maintained after 5 generations of passage (Fig. S3 and S4). Furthermore, the transcription level of T4SS genes in deletion and complementation mutants can also be confirmed by quantitative real-time PCR (qRT-PCR) (Fig. S5).

To test whether the mutagenesis affected bacterial growth, we compared the growth curves of Vp2S01::*cat*, Vp2S01Δ*traG*, Vp2S01Δ*trbE*, Vp2S01Δ*traG*::pRK415-*traG*, and Vp2S01Δ*trbE*::pRK415-*trbE*, which would serve as donor strains in the subsequent

**TABLE 1** Nucleotide and amino acid identity comparison and annotation of type IV secretion system between chromosome 2 of *Photobacterium profundum* SS9 and pVPGX1 of *Vibrio parahaemolyticus* 20130629002S01[a]

| Components | Coverage (nt) | E-value (nt) | Identity (nt) | Coverage (aa) | E-value (aa) | Identity (aa) | Predict function annotation |
|---|---|---|---|---|---|---|---|
| $TrbB_1$ | 76% | 2e-61 | 68.39% | 97% | 8e-144 | 60.78% | Conjugative transfer ATPase |
| $TrbB_2$ | 17% | 5e-18 | 80.23% | 97% | 9e-176 | 49.08% | Conjugative transfer ATPase |
| TrbC | 40% | 2e-20 | 73.33% | 96% | 3e-41 | 53.70% | A subunit of the pilus precursor |
| TrbD | 37% | 1e-07 | 68.03% | 88% | 2e-34 | 50.54% | Type IV secretory pathway, may provide energy for the export of DNA or the export of other Trb proteins |
| TrbE | 75% | 9e-139 | 68.06% | 99% | 0 | 63.24% | Conjugative transfer ATPase |
| TrbF | 13% | 0.001 | 76.32% | 99% | 1e-76 | 45.50% | Compose part of the pilus required for conjugative transfer |
| TrbG | 72% | 8e-47 | 66.93% | 96% | 5e-137 | 60.14% | A core component of the structure that forms the outer membrane pore |
| TrbH | 11% | 1e-11 | 83.67% | 100% | 2e-58 | 53.19% | A putative membrane lipoprotein lipid attachment site |
| TrbI | 43% | 3e-17 | 63.00% | 98% | 6e-104 | 42.86% | Not essential for conjugation but can greatly increase the conjugation efficiency |
| TrbJ | 13% | 4e-11 | 71.57% | 91% | 8e-78 | 45.89% | A core channel component |
| TrbL | 21% | 3e-37 | 76.10% | 89% | 3e-117 | 46.77% | Essential for plasmid through the inner membrane, probably a central component of the inner membrane channel |
| TrbN | 56% | 5e-19 | 67.87% | 100% | 3e-64 | 52.87% | Lytic transglycosylase domain-containing protein |
| TraF | / | / | / | 95% | 3e-50 | 42.33% | Conjugative transfer signal peptidase, can process the pilin subunit of the conjugative pilus into a circular form. |
| TraG | / | / | / | 9% | 0.009 | 30.00% | Coupling protein: link the conjugative transfer DNA-relaxase complex |

[a]"nt" means nucleotide; "aa" means amino acid; "/" means no significant similarity found.

conjugation assay. The results showed that the growth curve of any 2 strains was $P > 0.05$, indicating that the growth of these mutants was not affected (Fig. 3). This result ensured that the conjugative transfer would not be affected by the growth of different donors.

**The T4SS was able to mediate conjugative transfer.** We then carried out conjugation experiments with the above strains as donors and *Vc*LMB29 as a recipient at a ratio of 1:1. When *Vp*2S01Δ*trbE* was used as the donor strain, no positive clone was found from 213 selected clones. In the conjugation experiment of *Vp*2S01::*cat* and the *Vc*LMB29, 48 positive clones were detected from 216 selected clones on the 2216E-agar plate at the $10^{-1}$ dilution. The efficiency between the *Vp*2S01::*cat* and the *Vc*LMB29 was $(1.04 \pm 0.35) \times 10^{-8}$. When *Vp*2S01Δ*trbE*::pRK415-*trbE* was used as the donor strain, 10 positive clones were detected from 172 selected clones at the $10^{-1}$ dilution, and the efficiency was $(3.89 \pm 2.12) \times 10^{-9}$ (Fig. 4A).

Similarly, when *Vp*2S01Δ*traG* was used as the donor strain, no positive clones were found from 93 selected clones. In the conjugation experiment of *Vp*2S01::*cat* and the *Vc*LMB29, 28 positive clones were detected from 127 selected clones at the $10^{-1}$ dilution. The efficiency was $(6.44 \pm 3.77) \times 10^{-9}$. Moreover, when the complementation strain *Vp*2S01Δ*traG*::pRK415-*traG* was used as the donor strain, 5 positive clones were detected from 22 selected clones at the $10^{0}$ dilution, and 9 positive clones were detected from 50 selected clones at the $10^{-1}$ dilution. The conjugative transfer efficiency was $(1.59 \pm 0.92) \times 10^{-9}$ (Fig. 4B).

**Environmental factors could affect conjugation efficiency.** We further observed that different bacterial densities, temperatures, and nutrient levels could affect the efficiency of conjugative transfer (Fig. 5A). We changed the overall density of donor and recipient and carried out the conjugation experiment. The results showed that the conjugation efficiency was undetected with the bacterial densities at $10^6$ CFU/mL and $10^9$ CFU/mL. When the bacterial density was $10^{12}$ CFU/mL, the conjugative transfer

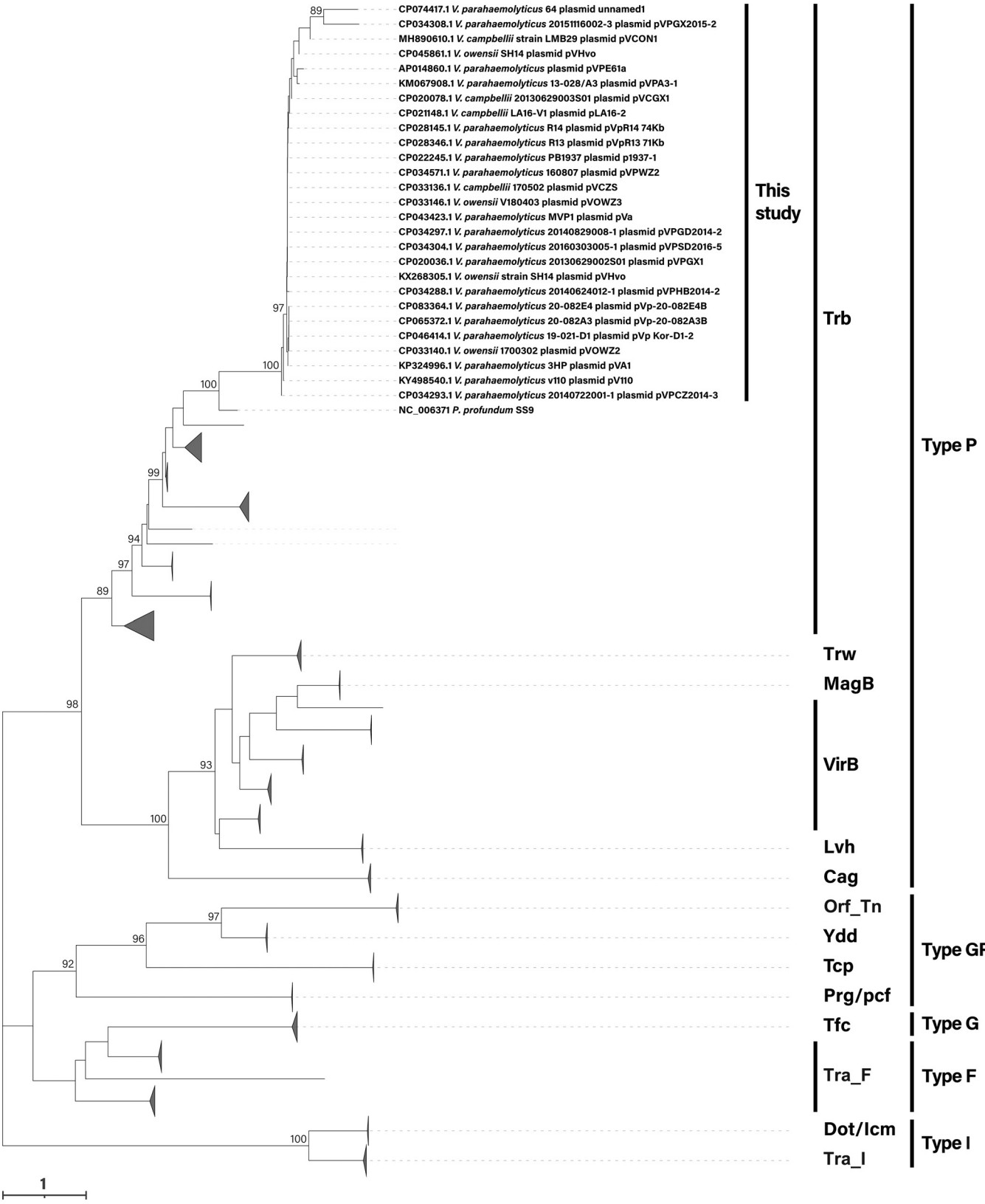

**FIG 2** Phylogenetic analyses of T4SS amino acid sequences. We used the amino acid sequences of TraG + TrbL+TrbE of the three components of Trb-type T4SS, and searched the corresponding components of the other types of T4SS. Percentage bootstrap values (1000 replicates) 85% are shown. GenBank accession numbers of the reference sequences are shown in Table S1. Scale bar represents the number of amino acid substitutions per site.

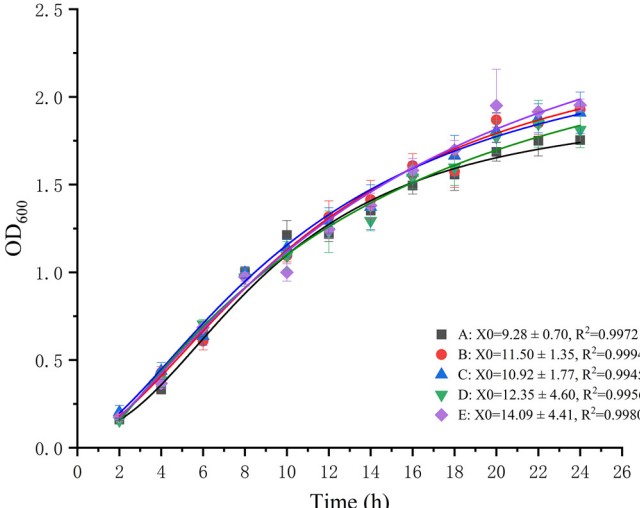

**FIG 3** Growth curves of the mutants. $R^2$: the square value of correlation coefficient of Logistic fitting curve. X0: the time when growth curve reaches maximum growth rate, and the result was presented as average value $\pm$ standard error (*SE*). (A) *Vp*2S01::*cat*, (B) *Vp*2S01$\Delta trbE$, (C) *Vp*2S01$\Delta traG$, (D) *Vp*2S01$\Delta trbE$::pRK415-*trbE*, (E) *Vp*2S01$\Delta traG$::pRK415-*traG*.

efficiency reached $(2.29 \pm 0.14) \times 10^{-8}$ (Fig. 5B). The conjugation experiments were also conducted at 5 temperature gradients. The results showed that no transconjugant was detected at 18, 23, and 38°C, whereas the conjugative transfer occurred at 28 and 34°C (Fig. 5C). As for different nutrient levels, conjugative transfer was not detected in 1/10 diluted 2216E broth, sterile seawater, or shrimp feed filtrate, but occurred in 2216E broth, LB broth, the filtrate of shrimp hepatopancreas, and M9 broth (Fig. 5D). These results showed that the temperature, bacterial density, and nutrient levels could affect the conjugation efficiency.

## DISCUSSION

Horizontal gene transfer plays a crucial role in bacterial adaptation, novel speciation, and evolution (36). Previous studies showed that AHPND pathogenic bacteria contain pVA1-type virulence plasmids. Non-pathogenic *Vibrio* bacteria can obtain virulence plasmids and become pathogenic through conjugative transfer (17). This mechanism has significantly diversified the causative agent and complicated the prevention strategies for AHPND. Therefore, it is necessary to understand the transfer mechanism of virulence plasmids to provide a theoretical basis for effective control of AHPND and to prevent diversification of the pathogenic *Vibrio* species.

Here, we found that all 27 reported pVA1-type plasmids have a T4SS associated with conjugation. The nucleotide and amino acid sequence identities of their Trb-type

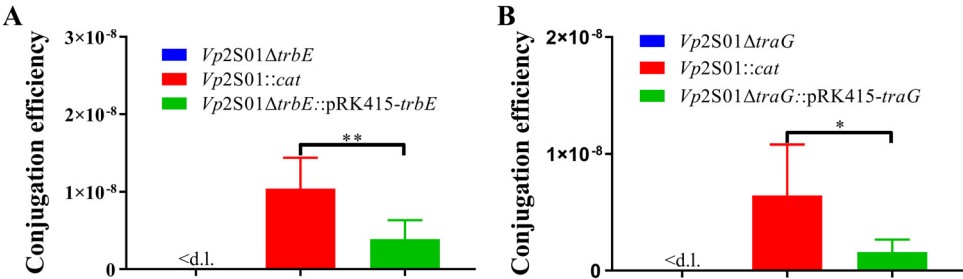

**FIG 4** The conjugation efficiency of the deletion strain, wild strain, and complementation strain. (A) The conjugation efficiency of the *trbE* gene deletion strain, wild strain, and complementation strains. (B) The conjugation efficiency of the *traG* gene deletion strain, wild strain, and complementation strain. *, $P < 0.05$, **, $P < 0.01$. <d.l.: below detection limit.

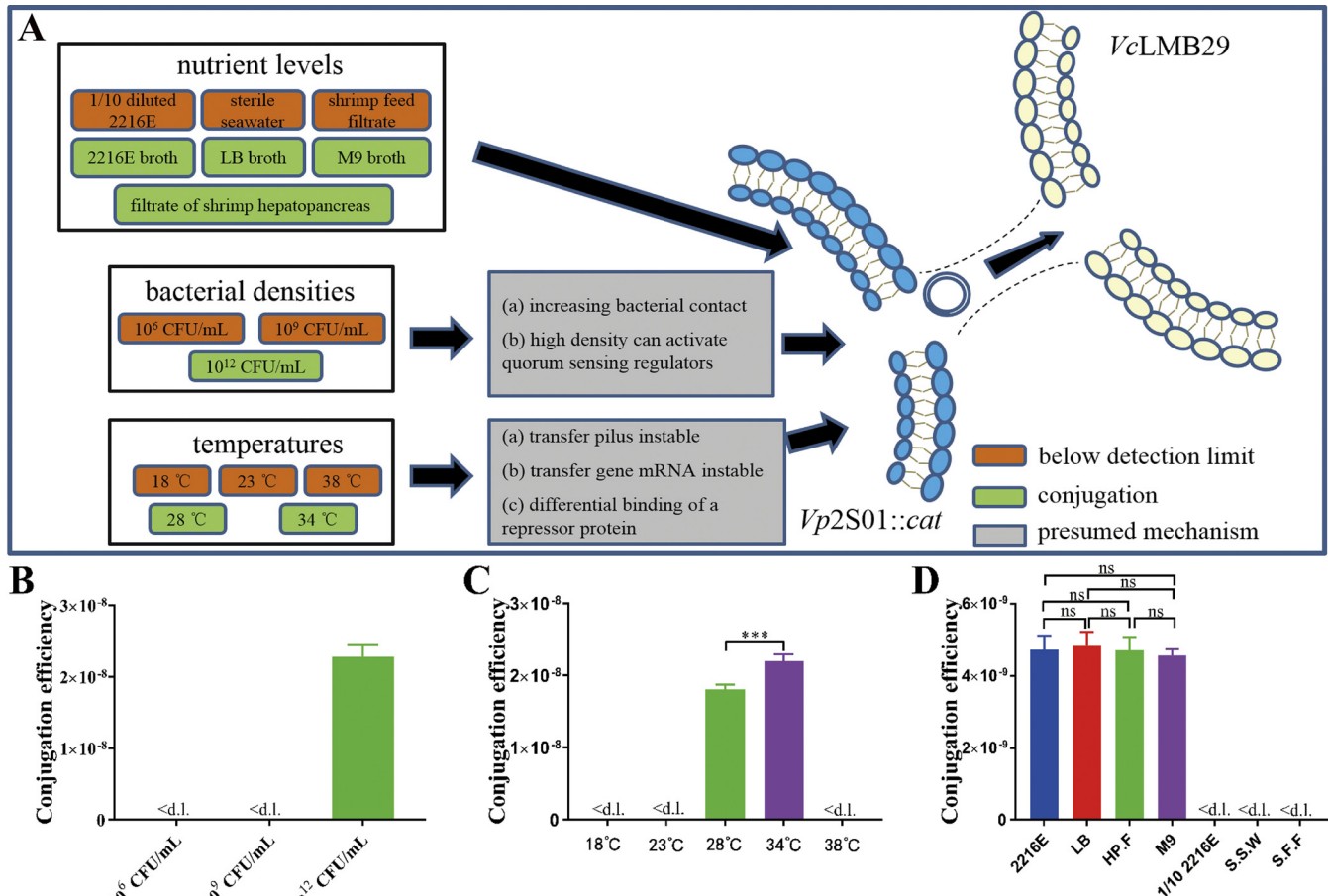

**FIG 5** Effects of different factors on conjugation efficiency. (A) Derivation graph of the influence of different factors on conjugative transfer. (B) The effect of different bacterial densities on conjugation efficiency. (C) The effect of different temperatures on conjugation efficiency. (D) The effect of different nutrient levels on conjugation efficiency. HP.F: the filtrate of shrimp hepatopancreas, S.S.W: sterile seawater, S.F.F: shrimp feed filtrate. ***, $P < 0.001$. ns: no significant difference. <d.l.: below detection limit.

T4SS systems are 89.54%–100.00% and 71.79%–100.00%, respectively (Fig. 1B and Fig. S2). These results showed that T4SSs in all pVA1-type plasmids were well conserved. Through sequence analysis and annotation, we identified genes that were important for conjugative transfer. Furthermore, we showed, through gene deletion and complementation analysis, that the T4SS could mediate the transfer of the pVA1-type plasmid. This paper reported the crucial components and functions of Trb-type T4SS in *Vibrio*, which provided a theoretical and experimental basis for future research on Trb-type T4SS. Although the Trb-type T4SS proteins of the pVA1-type plasmid have been predicted through NCBI and SecT4 websites, the actual function of those proteins remains to be verified. Furthermore, we predict the structure pattern of Trb-type T4SS based on the classical VirB-type T4SS (Fig. 1C). TrbH and TrbJ proteins are both core channel components in this Trb-type T4SS. However, those 2 components have no counterparts in the VirB-type T4SS. The structure in this novel Trb-type T4SS will require further study. On the other hand, although the transcription level of T4SS genes in deletion and complementation mutants can be confirmed by qRT-PCR (Fig. S5), the difference in the transcription level of T4SS genes of different strains is also worthy of being further studied.

Various environmental factors can affect conjugation efficiency. Our experimental results showed that temperature, bacterial density, and nutrient level influence the efficacy of the pVA1-type plasmid transfer (Fig. 5). The effects of temperature on conjugation have been well studied. Sherburne et al. reported that the R27 plasmid transfer from *Salmonella typhi* is temperature sensitive and is inhibited at and above 37°C (37).

In addition, conjugation of the Ti plasmid is also temperature sensitive (38). Liu et al. reported that the conjugation efficiency of pEIB202 plasmid in *Edwardsiella piscicida* is highest at 30°C, and its temperature ranges from 16°C to 37°C (39). However, there is little information on temperature-dependent regulation of T4SS in *Vibrio*. There are several possible temperature-regulated mechanisms for conjugation, such as pilus instability, mRNA instability of the transferring genes, or differential binding of (a) repressor protein(s) (Fig. 5A) (37). Thus, the regulatory mechanism of temperature needs further study. For bacterial density, we suspect there may be 2 reasons for the effect on conjugation efficiency. On the one hand, donor and recipient bacteria need to contact each other to form a mating pair formation to occur in conjugative transfer (19, 40, 41). Increasing bacterial density increases the probability of contact between them, which may increase the efficiency of conjugative transfer. On the other hand, the increase in bacterial density will increase the occurrence of quorum sensing (QS) among bacteria, and high density can activate QS regulators, which may activate the regulators of conjugative transfer and increase the efficiency of conjugative transfer (Fig. 5A) (42–45). Thus, the conjugative transfer could occur at $10^{12}$ CFU/mL, which did not detect at $10^6$ CFU/mL and $10^9$ CFU/mL. Although the bacterial density is difficult to reach $10^{12}$ CFU/mL in the natural environment, this might still occur in target organs or infected tissues in extreme cases. Different nutrient levels can affect the efficiency of conjugation efficiency. We compared conjugation efficiency using trophic media (including 2216E broth, LB broth, the filtrate of shrimp hepatopancreas, and M9 broth) and oligotrophic media (including 1/10 diluted 2216E broth, sterile seawater, and shrimp feed filtrate) (Fig. 5A and D) (46). The results showed that the conjugation efficiency was not detected with oligotrophic media but occurred with trophic media. Although the regulatory mechanism of these environmental factors is unknown, our results can explain the diversity of AHPND-causing *Vibrio* bacteria and identify the factors that can affect conjugation.

In conclusion, we demonstrated that the T4SS in pVA1-type plasmids is a novel Trb-type T4SS. We proved that it can mediate conjugative transfer of the pVA1-type plasmids and is regulated by environmental factors.

## MATERIALS AND METHODS

**Bioinformatic analysis of T4SS.** Homolog searches were performed using the NCBI BLAST tools (http://www.ncbi.nlm.nih.gov/blast). The identities of nucleotide and amino acid sequences were analyzed using BLASTn and BLASTp. The nucleotide sequence of the virulent plasmid pVPGX1 (GenBank: CP020036) of *V. parahaemolyticus* 20130629002S01 (*Vp*2S01) was used as a reference to obtain the high homologous plasmids by BLASTn. The T4SS components of these plasmids were identified online with the SecReT4 (https://bioinfo-mml.sjtu.edu.cn/SecReT4/index.php). According to positions of T4SS components on plasmids, the gene and CDS sequences of each T4SS component were downloaded from NCBI. Then, by BLASTn and BLASTp analysis of the T4SS sequence in pVPGX1 plasmid, the sequence with the highest homology was found, and its function was preliminarily predicted. Finally, taking *P. profundum* SS9 as a reference, the identity of amino acid sequence between pVPGX1 plasmid and *P. profundum* SS9 was compared by BLASTp. The conserved domains of the predicted polyprotein were analyzed using Pfam (47).

**Phylogenetic analyses of T4SS amino acid sequences.** We used the amino acid sequences of TraG + TrbL+TrbE of the 3 components of Trb-type T4SS, and searched the corresponding components of the other types of T4SS (Table S1). All types of T4SSs were combined in order for phylogenetic analysis. These sequences and amino acid sequences of the 27 pVA1-type plasmids were aligned by MAFFT (48) and sequences trimming is done with trimAl (49). A Neighbor-Joining (NJ) phylogenetic tree was constructed using MEGA 11.0.11 software (50). JTT+G was employed as the amino acid substitution model, and the phylogenetic tree was constructed by 1,000 bootstrap replicates. All sequences of the pVA1-type plasmids were downloaded from the GenBank database.

**Bacterial strains, plasmids, and culture conditions.** *Vp*2S01 harboring pVPGX1 was isolated from an AHPND-affected shrimp in China in 2013, and *V. campbellii* LMB29 (*Vc*LMB29) was isolated from a red drum (51, 52). *Vibrio* strains were cultured in marine 2216E broth (or agar) or tryptic soy broth with 2% NaCl (TSB+) at 28°C. *Escherichia coli* β2155 was cultured in Luria Bertani (LB) broth at 37°C, and it was also cultured in Brain Heart Infusion (BHI) Medium at 37°C. Bacterial strains and plasmids used in this study are shown in Table 2.

**Construction the *trbE* and *traG* gene deletion and complementation strains.** Deletion mutants *Vp*2S01Δ*trbE* and *Vp*2S01Δ*traG* were constructed with homologous recombination (17). We replaced the *trbE* gene on the pVPGX1 plasmid of the *Vp*2S01 strain with chloramphenicol resistant gene (chloramphenicol acetyltransferase, *cat*). In brief, the upstream and downstream homologous recombinant arms

**TABLE 2** Information of *Vibrio* strains and plasmids

| Strain/plasmid | Feature | Identification no. |
|---|---|---|
| Bacteria | | |
| *Vp*2S01 | Wild type carrying pVPGX1 | PMID: 29051747 |
| *Vp*2S01::*cat* | *Vp*2S01, chloramphenicol resistance gene (*cat*) inserted in pVPGX1 | PMID: 31231618 |
| *Vp*2S01Δ*trbE* | *Vp*2S01, the *trbE* in pVPGX1 replaced by *cat* | This study |
| *Vp*2S01Δ*traG* | *Vp*2S01, the *traG* in pVPGX1 replaced by *cat* | This study |
| *Vp*2S01Δ*trbE*::pRK415-*trbE* | *Vp*2S01Δ*trbE*, complemented with the *trbE* | This study |
| *Vp*2S01Δ*traG*::pRK415-*traG* | *Vp*2S01Δ*traG*, complemented with the *traG* | This study |
| *Vc*LMB29 | Wild type carrying rifampin resistant gene (*arr-9*) | PMID: 29109705 |
| *E. coli* β2155 | A diaminopimelic acid auxotrophic strain | Lab collection |
| | | |
| Plasmid | | |
| pVPGX1 | Wild type carrying *pirAB* | PMID: 29051747 |
| pCVD442 | Suicide plasmid carrying ampicillin resistant gene (*bla*) | Lab collection |
| pKD3 | Expression vector containing chloramphenicol resistance gene (*cat*) | Lab collection |
| pRK415 | Expression vector carrying tetracycline resistant gene (*tetA*) | Lab collection |
| pUC19 | Cloning vector carrying ampicillin resistant gene (*bla*) | Lab collection |

of the *trbE* gene and *cat* expression frame were amplified from the pVPGX1 and pKD3 plasmids, respectively, using primer pairs trbE-5F/5R, trbE-3F/3R and trbE-cat-F/R (Table 3). Three DNA fragments obtained were linked using gene splicing by overlap extension (SOE) PCR with the primer pair trbE-5F/3R and cloned into the pUC19. This fragment was subcloned into a suicide plasmid pCVD442 and electroporated into *E. coli* β2155, which was subsequently mated with *Vp*2S01 by conjugation. The transconjugant was inoculated on an LB plate containing chloramphenicol (10 μg/mL) and confirmed by PCR using the primer pairs trbE-out-F/R. Then, the positive bacterial colony of *Vp*2S01Δ*trbE* was inoculated on an LB plate containing chloramphenicol (17 μg/mL) and 10% sucrose and confirmed by PCR and Sanger sequencing using the primer pairs trbE-in-F/R and trbE-out-F/R (Table 3).

The plasmid pRK415 was used to construct complementation mutants. The *trbE* gene was amplified from *Vp*2S01 by PCR using the primer pair trbE-F/R (Table 3), then cloned into the corresponding site of pRK415 after digestion with HindIII and EcoRI. The complement plasmid pRK415-*trbE* was transformed into *E. coli* β2155. The bacterial colonies were screened by PCR using the primer pair pRK415-F/R (Table 3) on LB agar containing tetracycline (10 μg/mL). Then, the donor strain *E. coli* β2155/pRK415-*trbE* was mated with recipient strain *Vp*2S01Δ*trbE*. In combination with PCR using the primer pair pRK415-F/R (Table 3) and Sanger sequencing, the complementing transconjugant *Vp*2S01Δ*trbE*::pRK415-*trbE* was verified.

In the same way, we constructed the strains *Vp*2S01Δ*traG* and *Vp*2S01Δ*traG*:: pRK415-*traG*.

**Verification deletion and complementation strains.** To prove that the *trbE* gene deletion did not cause structural and functional damage to the adjacent genes, we verified the nucleotide and expression level of the genes adjacent to the *trbE* gene. The primer pairs trbE-del-1F/R and trbE-del-2F/R (Table 3) were used to verify the *trbD* gene on one side of the *trbE* gene. Similarly, we designed the primer pairs trbE-del-3F/R and trbE-del-4F/R (Table 3) to verify the *trbJ* gene on the other side of the *trbE* gene. The *Vp*2S01Δ*trbE* was cultured in 2216E broth for 10 h, and DNA/RNA were extracted from the pelleted bacteria. DNA was extracted by boiling at 95°C for 10 min; RNA was extracted using the RNAprep pure Cell/Bacteria Kit (Tiangen) and treated with DNase I at 25°C for 10 min. The extracted RNA was reverse transcribed into cDNA. DNase I was added during RNA extraction to ensure the complete digestion of genomic DNA. The cDNA was guaranteed to be produced by RNA reverse transcription.

To determine whether the *trbE* gene of the complementation strain *Vp*2S01Δ*trbE*::pRK415-*trbE* could be expressed, we continuously propagated it for 5 generations with 2216E broth and divided it into 2 groups: one group was cultured with 2 μg/mL tetracycline, and the other group was cultured without tetracycline. RNA and DNA were extracted from the first, the second, and the fifth generations, respectively (as shown in the above method), and the primer pairs trbE-com-F/R (Table 3) were used for PCR detection.

In addition, we used the same method to verify the *traG* gene deletion and complementation strains. We used the primer pairs, traG-del-1F/R, traG-del-2F/R, traG-com-F/R (Table 3) for PCR detection. The primer pairs traG-del-1F/R, traG-del-2F/R were used to identify the adjacent genes of deletion strain *Vp*2S01Δ*traG*. The primer traG-com-F/R was used to identify the adjacent genes of complementation strain *Vp*2S01Δ*traG*::pRK415-*traG*.

**Real-time quantitative PCR.** Samples of 5 donors, including *Vp*2S01::*cat*, *Vp*2S01Δ*trbE*, *Vp*2S01Δ*traG*, *Vp*2S01Δ*trbE*::pRK415-*trbE*, and *Vp*2S01Δ*traG*::pRK415-*traG* were collected. The total RNA per sample was treated with the RNA isolation kit (Tiangen) treated with DNase I following the manufacturer's instructions. First-strand cDNA was synthesized from 500 ng total RNA according to instructions given in SPARKscript II RT Plus Kit (SparkJade Biotech). The real-time quantitative (RT-qPCR) assay was performed in a total volume of 20 μL, containing 1 μL of cDNA, 0.4 μL of each primer (10 μM), 10 μL of 2 × SYBR green qPCR Mix, and 8.2 μL RNase-free water (SparkJade Biotech). The analysis was conducted with Bio-Rad CFX Opus 96 Real-Time PCR Instrument (Bio-Rad) under the following conditions: 94°C for 3 min; 40 cycles at 94°C for 10 s, and 60°C for 30 s. The gene-specific primers are listed in Table 3, and the relative expression of target genes was normalized by the $2^{-\Delta\Delta Ct}$ method with the housekeeping gene *gyrB* as an internal control.

**TABLE 3** The nucleotide sequences of PCR primers used in this study

| Primer | Sequence (5′–3′) |
|---|---|
| trbE-5F | ATAGTCGACTCATCTGGATTTGCGTTTGCATC |
| trbE-5R | GCTTGGTGGACTTGTCTTTGGATAAG |
| trbE-3F | CGTTGCGCTCTAGATTGGTGG |
| trbE-3R | ATAGTCGACTGCTGCCAATGCATCGAGTAC |
| trbE-cat-F | CTTATCCAAAGACAAGTCCACCAAGCGAGCTGCTTCGAAGTTCCTA |
| trbE-cat-R | CCACCAATCTAGAGCGCAACGCATATGAATATCCTCCTTAGTTCCTATTC |
| trbE-out-F | GAGCGGTTGAGTGAGCATCTTCT |
| trbE-out-R | CGAGCACCTACTATTAGGCGTCTC |
| trbE-in-F | GCTGCTCAATGTACTCACAGCAC |
| trbE-in-R | CTCAACCTGCATGTAGACGATGTG |
| pUC19-F | GTGCTGCAAGGCGATTAAGTT |
| pUC19-R | GCTCGTATGTTGTGTGGAATTG |
| traG-5F | ATAGTCGACGTTCAATCTTGCCTTGTAAAGCG |
| traG-5R | CAGAAACAGAAACTCTCGACACTAATGAAG |
| traG-3F | GATATTGATGCTCCTTAGTTGAATATCATCGG |
| traG-3R | ATAGTCGACGATGCACGAGGAGTGAGC |
| traG-cat-F | CTTCATTAGTGTCGAGAGTTTCTGTTTCTGGAGCTGCTTCGAAGTTCCTA |
| traG-cat-R | CCGATGATATTCAACTAAGGAGCATCAATATCCATATGAATATCCTCCT TAGTTCCTATTC |
| traG-out-F | CACTTGGTTGAGGTTTCGAGTGATCTG |
| traG-out-R | GTGTTGTCCTTTCGTGAACGTCTCG |
| traG-in-F | GATCGCGCATTTCAGCAAGCTG |
| traG-in-R | CGAAGAGATAATGGTGGCTGGATTCAG |
| pRK415-F | CAACGCAATTAATGTGAGTTAGCTCAC |
| pRK415-R | CTCTTCGCTATTACGCCAGCTG |
| trbE-F | TATAAGCTTGGAGCGCTTTTTTTTGCTTCTATGAATAGGGTTTTACTATG |
| trbE-R | TATGAATTCTCATTCGTCGTCCTCCAGGATGTGTTG |
| traG-F | ATATCTAGATTAACCCCATGCCTTGGAGCGTTCAGCGTAAAG |
| traG-R | ATAGAATTCGATGTAGCGAAGCCATGAGTGCCATTGTCTGGTG |
| trbE-del-1F | AGCAAAAAAAAGCGCTCGGG |
| trbE-del-1R | AGTGGTGTAAGGAGGCAAGG |
| trbE-del-2F | CCGCGCCCATGACTAAAAAG |
| trbE-del-2R | TGATCTTCGGTGGCGAGATG |
| trbE-del-3F | CTCGGTTCATCGTGACCTCC |
| trbE-del-3R | AGCGCACTGACCTTGTGCAG |
| trbE-del-4F | TTGAGCGGTTGAGTGAGCAT |
| trbE-del-4R | TTTGAGCAGTTGTTTGGCGG |
| trbE-com-F | ACATTCCACGCTGCCCAATA |
| trbE-com-R | TTGACCACGGTGACACCAAA |
| traG-del-1F | TCGTCCATTGGGTGGACAAC |
| traG-del-1R | AAAACGTGTCACGGTGCCTA |
| traG-del-2F | AAGCTGCGCTTTTTCAATGA |
| traG-del-2R | GTAAAGGCTTGGGCATCCAT |
| traG-com-F | GACATATTGCCTCTCGGCCA |
| traG-com-R | AAGGTGTCGCCTCAGGATTG |
| Vp-groelF | AGGTCAGGCTAAGCGCGTAAGC |
| Vp-groelR | GTCACCGTATTCACCCGTCGCT |
| Vca-hly5 | CTATTGGTGGAACGCAC |
| Vca-hly3 | GTATTCTGTCCATACAAAC |
| AP1F | CCTTGGGTGTGCTTAGAGGATG |
| AP1R | GCAAACTATCGCGCAGAACACC |
| VpPirB-392F | TGATGAAGTGATGGGTGCTC |
| VpPirB-392R | TGTAAGCGCCGTTTAACTCA |
| trbB$_1$-qRT-F | CGCATCATGGTGGGAGAAGT |
| trbB$_1$-qRT-R | GAATCGGCATGAATGGTGGC |
| trbB$_2$-qRT-F | GGCCGTTTAGTGGTGATGGA |
| trbB$_2$-qRT-R | TGCGCGATTTACAGTGGTCT |
| trbC-qRT-F | CCCTTAGAGCGCATTGTGGA |
| trbC-qRT-R | CATCTCGCCACCGAAGATCA |
| trbD-qRT-F | CTTTTTAGTCATGGGCGCGG |
| trbD-qRT-R | TAACTGCGCGTCGTTTTTGG |
| trbF-qRT-F | CGCAAATCTCACAGGCGTTC |

**TABLE 3** (Continued)

| Primer | Sequence (5′–3′) |
|---|---|
| trbF-qRT-R | TCGACCCATTCCATTTGCCA |
| trbG-qRT-F | CGGTTCGCGTCTACAACAAC |
| trbG-qRT-R | ACCACAAATTTCCCGTTGCG |
| trbH-qRT-F | CAGCTACGAGCCCGATCAAT |
| trbH-qRT-R | ATCGGACTGGCGAAGGAATG |
| trbI-qRT-F | CTCTTCGGTCAAGCCACCTT |
| trbI-qRT-R | ATCAGCAACGGACTCACCTG |
| trbJ-qRT-F | AAGCCTATCAGGGCCAATCG |
| trbJ-qRT-R | GCCAGTGCCAGTGAACTTTG |
| trbL-qRT-F | AGCAACAGTCGCTCTGTACC |
| trbL-qRT-R | ATGTTTTGCGTGGCTTGCTT |
| trbN-qRT-F | CTTTTGGGGTCAGCATTGCC |
| trbN-qRT-R | AGCGCAATGCTTGGTTTTGA |
| traF-qRT-F | CGCCTCCTAGTGCGGTTATT |
| traF-qRT-R | AAAAGAGAGATGGTCGCCCG |

**Bacterial growth rate.** To compare the difference in growth between the deletion and complementation strains, we measured the growth rates of the donor strains: Vp2S01::cat, Vp2S01ΔtrbE, Vp2S01ΔtraG, Vp2S01ΔtrbE::pRK415-trbE, Vp2S01ΔtraG::pRK415-traG. Each strain was cultured with 2216E broth, and then the bacterial suspension was diluted to $OD_{600}$ = 0.5. The 2 mL of bacterial suspension was added to 200 mL of 2216E broth. The inoculated broth was cultured in a shaker for 24 h at 28°C. The $OD_{600}$ value was used to represent the growth ability, and 3 parallels were made for each strain.

**Conjugation experiments with DNase I.** Conjugation experiments were carried out using a protocol described by Dong et al. (17). The donor and recipient in the experimental group were Vp2S01ΔtrbE and VcLMB29, respectively, whereas, in the control group, the donor was Vp2S01::cat. Briefly, the donor and recipient cells grew in 2216E supplemented with the corresponding antibiotics overnight at 28°C. For each mating assay, the donor and recipient cells were centrifuged, resuspended in 90 $\mu$L 2216E and 10 $\mu$L DNase I (20 unit; NEB), and placed onto a Millipore filter (0.22 $\mu$m pore size) layered a 2216E-agar plate. After 24 h of incubation at 28°C, the bacteria were washed from the filters in 2 mL of 2216E broth, diluted to $10^0$, $10^{-1}$, and $10^{-2}$. Then, 100 $\mu$L of the appropriate dilutions were plated on 2216E plates which contained chloramphenicol (60 $\mu$g/mL) and rifampin (20 $\mu$g/mL) as transconjugants selection antibiotics. In addition, by PCR, the primer pairs Vp-groelF/R, Vca-hly5/3 (53, 54), AP1F/R (55), and VpPirB-392F/R (56) (Table 3) were used to determine the positive transconjugants. The mixture was further diluted to $10^{-7}$, $10^{-8}$, and $10^{-9}$, then spread on 2216E plates containing rifampin (20 $\mu$g/mL) to determine the number of recipient cells. The conjugation efficiency was the average values ± standard deviations of the ratio of transconjugants' counts to those of recipient strains. The conjugative transfer of Vp2S01ΔtraG, Vp2S01ΔtrbE::pRK415-trbE or Vp2S01ΔtraG::pRK415-traG (donor), and VcLMB29 (recipient) was the same as Vp2S01ΔtrbE.

**Influence of different factors on conjugation efficiency.** In order to reveal the effect of different environmental factors on the conjugation efficiency of pVA1-type plasmid, we changed the bacterial density, temperature, and nutrient levels to carry out the conjugation experiment with Vp2S01::cat as the donor strain and VcLMB29 as the recipient strain. In the bacterial density experiment, other things being equal with conjugation experiments with DNase I section, we set up 3 final bacterial densities (donor:recipient = 1:1) at $10^6$, $10^9$, and $10^{12}$ CFU/mL, respectively. After 24 h of incubation at 28°C, the bacteria in each group were washed from the filters in 2 mL of 2216E broth, and the conjugation efficiency was the average value ± standard deviation (SD) of the ratio of transconjugants' counts to those of recipient strains. In the temperature change experiment, we set up five temperatures at 18, 23, 28, 34, and 38°C, respectively. Other things being equal, with conjugation experiments with DNase I section, the temperature of conjugation was changed to the temperatures described above, and the conjugation efficiency was calculated. In addition, other things being equal with conjugation experiments with DNase I section, we changed the nutrient levels of the conjugation experiment by replacing 2216E previously used in the conjugation experiment with 2216E broth, LB broth, the filtrate of shrimp hepatopancreas, M9 broth, 1/10 diluted 2216E broth, sterile seawater, and shrimp feed filtrate, respectively. The filtrate of shrimp hepatopancreas was extracted from shrimp hepatopancreas without AHPND. After 24 h of incubation at 28°C, the conjugation efficiency in each group was calculated.

**Statistical analysis.** The results of bacterial growth rate were calculated using the Logistic regression equation in the Origin tool. $R^2$: The square value of correlation coefficient of Logistic fitting curve. X0: The time when growth curve reaches maximum growth rate, and the result was presented as average value ± standard error (SE). A two-sample t test of the following formula was used to analyze the significant differences between growth curves (57).

$$T = \frac{X0_1 - X0_2}{\sqrt{SE_1{}^2 + SE_2{}^2}}$$

$P$ = T.DIST($T$, $n$-$2$, FALSE), where $n$ = 3. $P$ < 0.05 indicates significant difference in growth curve between the 2 strains.

The results of conjugation efficiency were used One-Way ANOVA in SPSS. A $P$ < 0.05 indicated that the difference was significant. A $P$ < 0.01 indicated a highly significant difference. A $P$ < 0.001 indicated an extremely significant difference.

## SUPPLEMENTAL MATERIAL

Supplemental material is available online only.

**SUPPLEMENTAL FILE 1**, PDF file, 0.7 MB.

## ACKNOWLEDGMENTS

We thank Zhe Zhao for the strain of *V. campbellii* LMB29.

This work was supported by the projects under the National Natural Science Foundation of China (31802342), National Key Research and Development Program of China (2018YFD0900501), Program for Chinese Outstanding Talents in Agricultural Scientific Research of the Ministry of Agriculture and Rural Affairs of the People's Republic of China, Central Public-interest Scientific Institution Basal Research Fund, CAFS (NO.2020TD39).

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
