## [Reviewer comments · Microbiology Spectrum]

Microbiology Spectrum

Conjugative transfer of AHPND-causing pVA1-type plasmid is mediated by a novel self-encoded type IV secretion system

Dehao Wang, Liying Wang, Dexi Bi, Jipeng Song, Guohao Wang, Ye Gao, Kathy Tang, Fanzeng Meng, Jingmei Xie, fan zhang, Jie Huang, Jianliang Li, and Xuan Dong

Corresponding Author(s): Xuan Dong, Yellow Sea Fisheries Research Institute, Chinese Academy of Fishery Sciences

Review Timeline:

Submission Date:	May 9, 2022
Editorial Decision:	June 10, 2022
Revision Received:	August 13, 2022
Accepted:	September 1, 2022

Editor: Jennifer Auchtung

Reviewer(s): Disclosure of reviewer identity is with reference to reviewer comments included in decision letter(s). The following individuals involved in review of your submission have agreed to reveal their identity: Songzhe Fu (Reviewer #2)

Transaction Report:

DOI: <https://doi.org/10.1128/spectrum.01702-22>

June 10, 2022

Dr. Xuan Dong
Yellow Sea Fisheries Research Institute of Chinese Academy of Fishery Sciences
Qingdao
China

Re: Spectrum01702-22 (Conjugative transfer of AHPND-causing pVA1-type plasmid is mediated by a novel self-encoded type IV secretion system)

Dear Dr. Xuan Dong:

Thank you for submitting your manuscript to Microbiology Spectrum. As you will see below, the reviewers identified several points within the manuscript where additional analyses are required before the manuscript can be considered. Please thoughtfully review and address these reviewer comments before submitting a revised manuscript. In addition, both reviewers indicated that English grammar of the final document needs work. ASM has prepared a website with helpful links to language editing services that should be considered during your revision. <https://journals.asm.org/language-editing-services>

Link Not Available

Sincerely,

Jennifer Auchtung

Journals Department
Reviewer comments:

Reviewer #1 (Comments for the Author):

In this manuscript, the authors report that a Trb-type T4SS was found in pVA1-type plasmid, and both the trbE gene, encoding an ATPase, and the traG gene that was annotated as a type IV coupling protein (T4CP) were characterized as key components of the T4SS. They also observed that factors of bacterial density, temperature and nutrient levels appeared to regulate the conjugation efficiency. These results contribute to better understanding of the mechanisms involved in horizontal transfer of the pVA1-type plasmid.

Major concerns:

1. As shown in Figure 4, the conjugation efficiency was significantly differed in between the control strains and complementation strains, even though the complementation strains were observed with the restored conjugation capability in comparison to the mutant strains. It indicates that the deletion of *trbE* and *traG* likely affected the expression of the other genes of the T4SS. Accordingly, I suggest the authors to perform RT-qPCR to analyze the expression profiles of these genes shown in Table 1.
2. The conjugants were undetectable upon the knockout of the *trbE* and *traG*. Notably, the data presented in their study failed to exclude the possibility that the mutant plasmid was able to transfer to the receptor strain but unable to replicate itself in it. I suggest the authors to perform electroporation to further address this issue.
3. The resolution of Figure 2 is too low to be analyzed.
4. Please provide detail methods used in analyzing of "Environmental factors could affect conjugation", e.g., cell age, culture medium, growth condition.
5. Line 115 and Table 1, I do not understand why there is 0% sequence identity between gene counterparts.

Reviewer #2 (Comments for the Author):

The pathogenic pVA1-type plasmids that carry *pirAB* toxin genes are genetic basis for *Vibrio* to cause acute hepatopancreatic necrosis disease (AHPND). The manuscript entitled "Conjugative transfer of AHPND-causing pVA1-type plasmid is mediated by a novel self-encoded type IV secretion system" is an interesting read. Authors experimentally confirmed that the T4SS was able to mediate the conjugation of

27 pVA1-type plasmids, which improved our understanding about the role of T4SS in promoting conjugative transfer of AHPND-causing pVA1-type plasmid. However, there is a fair bit of clarifications need to be done to link the data with interpretations, which would potentially increase the chance of the acceptance.

Major comments:

1. As authors emphasized that the T4SS in pVA1-type plasmids is a novel *Trb*-type member, a detailed phyogenic and evolutionary analysis is needed. The statement in this section seems pay more attention to the function prediction of individual genes of T4SS. More in-depth phylogenetic analysis is needed, which is critical and missing. Authors can have a look at another similar work as reference (<https://www.frontiersin.org/articles/10.3389/fmicb.2022.853744/full>).
2. Control the conjugative transfer of pVA1-type plasmids is an important mean for controlling AHPND, which is not addressed in the discussion section.
An example can be referred: <https://doi.org/10.1128/microbiolspec.MTBP-0015-2016>
3. There are some issues for experimental design. For instance, nutrient level can not be quantified.

Detail comments

Introduction:

the introduction section is well organized. However, it worths mentioned the gaps author wish to fill in the introduction as well as discussion section.

line 113: the full name of *P. profundum* needs to mention at first time.

Results

1. The T4SS in pVA1-type plasmids is a novel *Trb*-type member

Line 91: Reviewer suggests that the author clarify the definition of pVPGX1 type plasmid in this study and its relationship with pVA1 type plasmid in line117, also for Line 168 "all 26 reported pVA1-type plasmids"

Line 111-117: The phylogenetic analysis of *trbE* alone is not enough to draw the conclusion that T4SS in pVA1 type plasmas is a novel *Trb* type member

(1) Please explain the considerations for sequence selection: why only *trbE* gene sequence is selected for phylogenetic analysis, and why *P. profundum* SS9 strain is selected as the reference sequence

(2) Please explain the current T4SS classification and its classification standard

To prove it is a novel *Trb* type member, we should include at least the following evidences:

(1) The phylogenetic relationships of major *Trb* genes in pva1 and all related plasmids are clustered together

(2) The representative sequences of different members of the existing T4SS are included in the phylogenetic tree and T4SS from pVA1 plasmids are clearly distinguished from the above sequences and fall into a separate branch

2. Construction of the deletion and complementation mutants of *trbE* and *traG*

Line 118: why only *TrbE* and *TraG* were selected

Line 122-123: typo: "Successful deletions were confirmed PCR."

Line 123-124: qPCR data is needed here to prove the adjacent genes remained intact and could be expressed unaffectedly

Line 126: what is the meaning of "Normal expression"

3. The T4SS was able to mediate conjugative transfer

Line136: reviewer can see that *TraB* and *TraG* play an important role in constructive transfer, but it is difficult to prove that the

whole T4SS system "was able to mediate constructive transfer"

Line 140-141 the specific data of "transconjugant/recipient" and "untatable" is needed. Need to indicate how many clones were picked, and how many were positive.

4. Environmental factors could affect conjugation efficiency

Line 150-151: in natural water, it is difficult to imagine that the bacterial concentration reaches 10^{12} cfu/ml. Please discuss it reasonably

Line 153: is there any reasonable consideration for the author to select 18, 23, and 38 °C as the culture temperature? (especially 38 °C is not the conventional breeding temperature)

line 155-156: The significance of nutrient levels is unclear. It is suggested that the author measured the nutrient levels by the level of carbon, nitrogen and other key indicators that can quantify the nutrient level of the culture medium.

In addition, in the experimental design, did the author consider the influence of dilution on the salinity and osmotic pressure of the medium in the process of diluting the medium?

Discussion

In Fig. 1, please discuss the deletion of TrbH and J homologous sequences.

Line 179-180: "unknown functional genes in Trb type T4SS gene clusters were required for further study." Some specific discussion is needed here.

Line 189-192: no experimental data or published literature support this statement.

Line 178-179: language issue: "whether its actual function is the same as predicted remains to be verified."

Line 209: The experimental data of this study are quite different from the actual aquaculture environment, so it is difficult to say that it has practical guiding significance for the prevention and control of AHPND.

Line 212-213 "We proved that it can mediate conjugative transfer of the pVA1-type plasmids and is regulated by environmental factors." This conclusion is not solid.

Materials and methods

Line 329-345 There are some issues for experimental design

bacterial densities :

Need to give the reasons why selecting three initial density levels and the reason why kept donor:recipient=1:1, as the ratio of donor and acceptor are not consistent with the natural environment.

The final bacterial concentration of each experimental group is not given. Also, conjugation experimnt in liquid culture might be more meaningful.

temperature change :

Need to indicate the temperature fluctuation range in the actual experiment.

Since the final bacterial concentration at the end of the experiment is not given, please explain how to rule out the possibility that different culture temperatures lead to different bacterial densities, which may affect the conjugation efficiency.

nutrient level

Lack of quantifiable indicators, such as calorie, protein, carbohydrate, fat, etc (shrimp feed filter, the composition of feed shall be indicated)

Another question: the author specially emphasized 2% NaCl in this section. In the dilution experiment, are the salinities of various media the same? Is osmotic pressure the same?

Line 237 why use LG+G+I+F model? needs explanation.

Line 307 "Inoculated broth was cultured in a shaker for 24 h." temperature is need to specified.

Table 1: only identity value , coverage and E-value are needed

Resolution for Fig.2 is low. To exhibit each branch, a traditional tree is needed instead of circle tree.

Staff Comments:

Preparing Revision Guidelines

- Point-by-point responses to the issues raised by the reviewers in a file named "Response to Reviewers," NOT IN YOUR COVER LETTER.
- Upload a compare copy of the manuscript (without figures) as a "Marked-Up Manuscript" file.

- Each figure must be uploaded as a separate file, and any multipanel figures must be assembled into one file.
- Manuscript: A .DOC version of the revised manuscript
- Figures: Editable, high-resolution, individual figure files are required at revision, TIFF or EPS files are preferred

Please return the manuscript within 60 days; if you cannot complete the modification within this time period, please contact me. If you do not wish to modify the manuscript and prefer to submit it to another journal, please notify me of your decision immediately so that the manuscript may be formally withdrawn from consideration by Microbiology Spectrum.

Dear Editor and Reviewers,

We would like to thank you for your comments on our manuscript entitled “Conjugative transfer of AHPND-causing pVA1-type plasmid is mediated by a novel self-encoded type IV secretion system” (ID: Spectrum01702-22). The comments are thoughtful and in-depth, and definitely helped us to improve our manuscript. We have now revised the manuscript according to your comments. We have made point-by-point response to each question/comment raised by you.

Microbiology Spectrum

Manuscript ID : Spectrum01702-22

Title: Conjugative transfer of AHPND-causing pVA1-type plasmid is mediated by a novel self-encoded type IV secretion system

Reviewer: #1

In this manuscript, the authors report that a Trb-type T4SS was found in pVA1-type plasmid, and both the *trbE* gene, encoding an ATPase, and the *traG* gene that was annotated as a type IV coupling protein (T4CP) were characterized as key components of the T4SS. They also observed that factors of bacterial density, temperature and nutrient levels appeared to regulate the conjugation efficiency. These results contribute to better understanding of the mechanisms involved in horizontal transfer of the pVA1-type plasmid.

Major concerns:

1. As shown in Figure 4, the conjugation efficiency was significantly differed in between the control strains and complementation strains, even though the complementation strains were observed with the restored conjugation capability in comparison to the mutant strains. It indicates that the deletion of *trbE* and *traG* likely affected the expression of the other genes of the T4SS. Accordingly, I suggest the authors to perform RT-qPCR to analyze the expression profiles of these genes shown in Table 1.

Response: Thanks for your suggestion and we have carried out the quantitative real-time PCR based on your suggestion. We have added this information in RESULTS (Lines 132-133) and MATERIALS AND METHODS sections (Lines 323-336).

2. The conjugants were undetectable upon the knockout of the *trbE* and *traG*. Notably, the data presented in their study failed to exclude the possibility that the mutant plasmid was able to transfer to the receptor strain but unable to replicate itself in it. I suggest the authors to perform electroporation to further address this issue.

Response: Thanks for your suggestion. In our study, the pRK415 plasmid, which was cloned into the *trbE/traG* gene, was used as the complementary plasmid. The complementation plasmid was transformed into the deletion strain, and the complementation strain *Vp2S01ΔtrbE::pRK415-trbE/Vp2S01ΔtraG::pRK415-traG* was successfully constructed. Therefore, the complementation strain and the deletion strain harbor the same gene mutation plasmid, and the complementation strain can carry out normal gene replication and conjugative transfer. Therefore, the possibility that the mutant plasmid can be transferred to the receptor strain but cannot be replicated was excluded.

3. The resolution of Figure 2 is too low to be analyzed.

Response: We have redrawn Figure 2 and resubmitted.

4. Please provide detail methods used in analyzing of "Environmental factors could affect conjugation", e.g., cell age, culture medium, growth condition.

Response: Thank you for your suggestion that we have revised the "Influence of different factors on conjugation efficiency" section. We revised "In order to reveal the effect of different factors on the conjugation efficiency of pVA1-type plasmid, we changed the bacterial density, temperature, and nutrient levels to carry out the conjugative transfer experiment with *Vp2S01::cat* as the donor strain and *VcLMB29* as the recipient strain. In the conjugative transfer experiment, we set up three bacterial densities (donor:recipient=1:1) at 10^6 , 10^9 , and 10^{12} CFU/mL, respectively. Under the same conditions, the experiments were carried out for 24 h, respectively. In the external temperature change experiment, we set up five temperatures at 18, 23, 28, 34, and 38°C, respectively. The temperature of conjugative transfer in the previous experiment was changed to the temperatures describe above, and the conjugative transfer experiments for 24 h were carried out, respectively. In addition, we changed the nutrient levels of the conjugative transfer experiment by replacing 90 μ L 2216E and 10 μ L DNase I previously used in the conjugative transfer experiment with 2216E broth, LB broth, the filtrate of shrimp hepatopancreas, M9 broth, 1/10 diluted 2216E broth, sterile seawater and shrimp feed filtrate. Conjugative transfer occurs on sterile plate. The filtrate of shrimp hepatopancreas was extracted from shrimp hepatopancreas without AHPND." to "In order to reveal the effect of different environmental factors on the conjugation efficiency of pVA1-type plasmid, we changed the bacterial density, temperature, and nutrient levels to carry out the conjugation experiment with *Vp2S01::cat* as the donor strain and *VcLMB29* as the recipient strain. In the bacterial density experiment, other things being equal with conjugation experiments with DNase I section, we set up three final bacterial densities (donor:recipient = 1:1) at 10^6 , 10^9 , and 10^{12} CFU/mL, respectively. After 24 h incubation at 28°C, the bacteria in each group were washed from the

filters in 2 mL of 2216E broth, and the conjugation efficiency was the average values \pm standard deviations of the ratio of transconjugants' counts to those of recipient strains. In the temperature change experiment, we set up five temperatures at 18, 23, 28, 34, and 38°C, respectively. Other things being equal, with conjugation experiments with DNase I section, the temperature of conjugation was changed to the temperatures described above, and the conjugation efficiency was calculated. In addition, other things being equal with conjugation experiments with DNase I section, we changed the nutrient levels of the conjugation experiment by replacing 2216E previously used in the conjugation experiment with 2216E broth, LB broth, the filtrate of shrimp hepatopancreas, M9 broth, 1/10 diluted 2216E broth, sterile seawater, and shrimp feed filtrate, respectively. The filtrate of shrimp hepatopancreas was extracted from shrimp hepatopancreas without AHPND. After 24 h incubation at 28°C, the conjugation efficiency in each group was calculated.” (Line 365-385).

5. Line 115 and Table 1, I do not understand why there is 0% sequence identity between gene counterparts.

Response: Thanks for your comments. We have revised “0%” to “/” in Table 1, which means no similarity found.

Reviewer: #2

The pathogenic pVA1-type plasmids that carry pirAB toxin genes are genetic basis for *Vibrio* to cause acute hepatopancreatic necrosis disease (AHPND). The manuscript entitled "Conjugative transfer of AHPND-causing pVA1-type plasmid is mediated by a novel self-encoded type IV secretion system" is an interesting read. Authors experimentally confirmed that the T4SS was able to mediate the conjugation of 27 pVA1-type plasmids, which improved our understanding about the role of

T4SS in promoting conjugative transfer of AHPND-causing pVA1-type plasmid. However, there is a fair bit of clarifications need to be done to link the data with interpretations, which would potentially increase the chance of the acceptance.

Major comments:

1. As authors emphasized that the T4SS in pVA1-type plasmids is a novel Trb-type member, a detailed phylogenetic and evolutionary analysis is needed. The statement in this section seems pay more attention to the function prediction of individual genes of T4SS. More in-depth phylogenetic analysis is needed, which is critical and missing. Authors can have a look at another similar work as reference.

Response: Thank you for your suggestion. The reason that we initial chose the ATPase component to construct phylogenetic tree was because it has been proven to be an efficient component to classify T4SS (see the reference below). Here, in order to make more in-depth phylogenetic analysis, we have now performed a new phylogenetic analysis from the amino acid sequences of TraG+TrbL+TrbE components.

Reference: Bi D, Liu L, Tai C, Deng Z, Rajakumar K, Ou H. 2013. SecReT4: a web-based bacterial type IV secretion system resource. *Nucleic Acids Res* 41:D660-5. <https://doi.org/10.1093/nar/gks1248>.

2. Control the conjugative transfer of pVA1-type plasmids is an important mean for controlling AHPND, which is not addressed in the discussion section.

Response: Thanks for your suggestion and we have added to the

DISCUSSION section (Line 179-181 and 201-232).

3. There are some issues for experimental design. For instance, nutrient level can not be quantified.

Response : Actually, we meant to evaluate the risk of conjugative transfer in different survival factors. Sterile seawater and 2216E broth are used to simulate seawater condition. 1/10 diluted 2216E broth is used to simulate desalination condition. Shrimp feed filtrate is used to simulate the aquaculture condition after feeding. The filtrate of shrimp hepatopancreas is used to simulate the condition inside the host. LB and M9 broth are used to simulate the lab condition. As for the quantifiable indicators, it can be further studied in the future work.

Detail comments

Introduction:

the introduction section is well organized. However, it worths mentioned the gaps author wish to fill in the introduction as well as discussion section.

line 113: the full name of *P. profundum* needs to mention at first time.

Response: Thank you for your valuable suggestion and we have added the full name of *P. profundum* at the first time (Line 116).

Results

1. The T4SS in pVA1-type plasmids is a novel Trb-type member.

Line 91: Reviewer suggests that the author clarify the definition of pVPGX1 type plasmid in this study and its relationship with pVA1-type plasmid in line 117, also for Line 168 "all 26 reported pVA1-type plasmids".

Response: Thanks for your comment and sorry for our mistake. pVPGX1 plasmid is one of the pVA1-type plasmids. We revised “pVPGX1-type plasmids” to “pVA1-type plasmids” (Line 96).

Line 111-117: The phylogenetic analysis of *trbE* alone is not enough to draw the conclusion that T4SS in pVA1 type plasmas is a novel Trb type member

(1) Please explain the considerations for sequence selection: why only *trbE* gene sequence is selected for phylogenetic analysis, and why *P. profundum* SS9 strain is selected as the reference sequence.

(2) Please explain the current T4SS classification and its classification standard. To prove it is a novel Trb type member, we should include at least the following evidences:

(1) The phylogenetic relationships of major Trb genes in *pva1* and all related plasmids are clustered together

(2) The representative sequences of different members of the existing T4SS are included in the phylogenetic tree and T4SS from pVA1 plasmids are clearly distinguished from the above sequences and fall into a separate branch.

Response: Thank you for your suggestion. As we mentioned above, we used ATPase component to construct phylogenetic tree based on the following reference. We have performed a new phylogenetic analysis from the amino acid sequences of TraG+TrbL+TrbE components based on your suggestion. The *P. profundum* SS9 strain was used as a reference sequence because the results of our phylogenetic analysis showed that the T4SS we studied was the closest to the clade in which it was located and was more relevant for reference.

Reference: Bi D, Liu L, Tai C, Deng Z, Rajakumar K, Ou H. 2013. SecReT4: a web-based bacterial type IV secretion system resource. *Nucleic Acids Res* 41:D660-5. <https://doi.org/10.1093/nar/gks1248>.

2. Construction of the deletion and complementation mutants of *trbE* and *traG*.

Line 118: why only TrbE and TraG were selected.

Response: In this study, we identified T4SS's essential components including a *traG* as a T4CP gene and a *trbE* gene as an ATPase gene. Construction of deletion and complementation mutants of the two essential components (*trbE* and *traG*) proved that the T4SS mediate the conjugative transfer of pVA1-type plasmids. The TraG protein is a type IV coupling protein (T4CP), which can connect the conjugative transfer DNA-relaxase complex (Kwak, et al., 2017; de la Cruz, et al., 2010). The *trbE* gene is encoded an ATPase, which is the essential components in the conjugation process. We have introduced it in INTRODUCTION section (Line 80-86).

Kwak MJ, Kim JD, Kim H, Kim C, Bowman JW, Kim S, Joo K, Lee J, Jin KS, Kim YG. 2017. Architecture of the type IV coupling protein complex of *Legionella pneumophila*. *Nat Microbiol* 2:17114.

de la Cruz F, Frost LS, Meyer RJ, Zechner EL. 2010. Conjugative DNA metabolism in Gram-negative bacteria. *FEMS Microbiol Rev* 34:18-40.

Line 122-123: typo: "Successful deletions were confirmed PCR."

Response: Thanks for your comments and we have revised "Successful deletions were confirmed PCR" to "Deletions of these two genes were successfully confirmed using PCR and Sanger sequencing" (Lines 125-126).

Line 123-124: qPCR data is needed here to prove the adjacent genes remained intact and could be expressed unaffectedly.

Response: Thanks for your suggestion and we have added the qPCR results in this study (Line 132-133 and Fig S5).

Line 126: what is the meaning of "Normal expression".

Response: Thanks for your kind reminder and we have revised "Normal expression" to "Transcription" (Line 129).

3. The T4SS was able to mediate conjugative transfer.

Line136: reviewer can see that TraB and TraG play an important role in constructive transfer, but it is difficult to prove that the whole T4SS system "was able to mediate constructive transfer".

Response: In this study, we identified the T4SS system in pVA1-type plasmids and its essential components including a *traG* as a T4CP gene and a *trbE* gene as an ATPase gene. Construction of deletion and complementation mutants of the *trbE* and *traG* proved that the T4SS

mediate the conjugative transfer of pVA1-type plasmids.

Line 140-141 the specific data of "transconjugant/recipient" and "untatable" is needed. Need to indicate how many clones were picked, and how many were positive.

Response: Thank you for your suggestion and we have revised to “We then carried conjugation experiments with the above strains as donors and VcLMB29 as a recipient at a ratio of 1:1. When *Vp2S01ΔtrbE* was used as the donor strain, no positive clone was found from 213 selected clones. In the conjugation experiment of *Vp2S01::cat* and the VcLMB29, 48 positive clones were detected from 216 selected clones on the 2216E-agar plate at 10^{-1} dilution of. The efficiency between the *Vp2S01::cat* and the VcLMB29 was $(1.04 \pm 0.35) \times 10^{-8}$. When *Vp2S01ΔtrbE::pRK415-trbE* was used as the donor strain, 10 positive clones were detected from 172 selected clones at 10^{-1} dilution and the efficiency was $(3.89 \pm 2.12) \times 10^{-9}$ (Fig. 4A).

Similarly, when *Vp2S01ΔtraG* was used as the donor strain, no positive clones were found from 93 selected clones. In the conjugation experiment of *Vp2S01::cat* and the VcLMB29, 28 positive clones were detected from 127 selected clones at 10^{-1} dilution. The efficiency was $(6.44 \pm 3.77) \times 10^{-9}$. Moreover, when the complementation strain *Vp2S01ΔtraG::pRK415-traG* was used as the donor strain, 5 positive clones were detected from 22 selected clones at 10^0 dilution and 9 positive clones were detected from 50 selected clones at 10^{-1} dilution. The conjugative transfer efficiency was $(1.59 \pm 0.92) \times 10^{-9}$ (Fig. 4B).”

(Lines 142-158).

4. Environmental factors could affect conjugation efficiency.

Line 150-151: in natural water, it is difficult to imagine that the bacterial concentration reaches 10^{12} cfu/ml. Please discuss it reasonably.

Response : Thanks for your suggestion. The aim of our study is to focus more on the conjugation of pVA1-type plasmids across *Vibrio spp.* is mediated by a novel T4SS and regulated by environmental factors. Although our study has low fitness for simulated production, the results show that the increase in bacterial density will lead to an increase in the species of bacterial causing AHPND, which is unfavorable for the prevention and treatment of the disease. We have add “Although the bacterial density is difficult to reach 10^{12} CFU/mL in the natural environment, this might still occur in target organs or infected tissues in extreme cases.” to the DISCUSSION section (Lines 222-224).

Line 153: is there any reasonable consideration for the author to select 18, 23, and 38°C as the culture temperature? (especially 38°C is not the conventional breeding temperature).

Response : (1) 18-38°C is the normal growth temperature of *Vibrio*. In this temperature range, we set up five temperature gradient experiments to find the appropriate temperature range of conjugation, which provides the basis for future research. (2) Although 38°C is not the conventional breeding temperature. *Vibrio* can cause human diseases, such as cholera

and gastroenteritis, while the body temperature is about 36.8°C. This study can also provide reference for the conjugation of other *Vibrio*.

line 155-156: The significance of nutrient levels is unclear. It is suggested that the author measured the nutrient levels by the level of carbon, nitrogen and other key indicators that can quantify the nutrient level of the culture medium.

Response : Actually, we want to evaluate the risk of conjugative transfer in different survival factors. Sterile seawater and 2216E broth are used to simulate seawater condition. 1/10 diluted 2216E broth is used to simulate desalination condition. Shrimp feed filtrate is used to simulate the aquaculture condition after feeding. The filtrate of shrimp hepatopancreas is used to simulate the condition inside the host. LB and M9 broth are used to simulate the lab condition.

In addition, in the experimental design, did the author consider the influence of dilution on the salinity and osmotic pressure of the medium in the process of diluting the medium?

Response: The salinity and osmotic pressure of 2216E, LB and M9 were also different, and all of them could undergo conjugative transfer. Therefore, the effects of salinity and osmotic pressure on conjugation were relatively small in this experiment.

Discussion

In Fig. 1, please discuss the deletion of TrbH and J homologous sequences.

Response: Thank you for your valuable suggestion and we have made modifications (Line 194-196).

Line 179-180: "unknown functional genes in Trb type T4SS gene clusters were required for further study." Some specific discussion is needed here.

Response: Thank you for your valuable suggestion and we have made modifications (Line 189-200).

Line 189-192: no experimental data or published literature support this statement.

Response : We found the paper to support our statement and we have added it to reference (Line 208-212).

Sherburne CK, Lawley TD, Gilmour MW, Blattner FR, Burland V, Grotbeck E, Rose DJ, Taylor DE. 2000. The complete DNA sequence and analysis of R27, a large IncHI plasmid from *Salmonella typhi* that is temperature sensitive for transfer. *Nucleic Acids Res* 28:2177-2186. <https://doi.org/10.1093/nar/28.10.2177>.

Line 178-179: language issue:"whether its actual function is the same as predicted remains to be verified."

Response : Thank you for your reminder and we have revised “whether its actual function is the same as predicted remains to be verified” to “the actual function of those proteins remains to be verified” (Lines 192-193).

Line 209: The experimental data of this study are quite different from the actual aquaculture environment, so it is difficult to say that it has practical guiding significance for the prevention and control of AHPND.

Response : Thank you for your valuable suggestion. Actually, bacterial densities, temperatures and nutrient levels are the essential factor for shrimp aquaculture. We designed this assay to provide a theoretical basis for effective control of AHPND and to prevent diversification of the pathogenic *Vibrio* species. In order to be precise, we revised “form the basis of management strategies leading to the prevention and control of AHPND” to “identify the factors that can affect conjugation”. (Lines 231-232)

Line 212-213" We proved that it can mediate conjugative transfer of the pV A1-type plasmids and is regulated by environmental factors." This conclusion is not solid.

Response : In this study, we experimentally confirmed that the T4SS was able to mediate the conjugation of pVA1-type plasmids. A *trbE* gene encoding an ATPase and a *traG* gene annotated as a type IV coupling protein (T4CP) were characterized as key components of the T4SS. Deletion of either gene abolished the conjugative transfer of a pVA1-type plasmid from AHPND causing *V. parahaemolyticus* to *V. campbellii*, which was restored by complementation of the corresponding gene. Moreover, we found that bacterial density, temperature and nutrient levels are factors that can regulate the conjugation efficiency. In conclusion, we proved that the conjugation of pVA1-type plasmids across *Vibrio* spp. is mediated by a novel T4SS and regulated by environmental factors.

Materials and methods

Line 329-345 There are some issues for experimental design.

bacterial densities :

Need to give the reasons why selecting three initial density levels and the reason why kept donor:recipient=1:1, as the ratio of donor and acceptor are not consistent with the natural environment.

Response : Actually, it's hard to decide the ratio of donor and recipient consistent with the natural environment. In our previous study, we have established the conjugation model of pVA1-type plasmid. In this study, we use the same model to evaluate the conjugation efficiency in different densities.

Reference: Dong X, Song J, Chen J, Bi D, Wang W, Ren Y, Wang H, Wang G, Tang K, Wang X, Huang J. 2019. Conjugative Transfer of the pVA1-Type Plasmid Carrying the *pirAB* (vp) Genes Results in the Formation of New AHPND-Causing *Vibrio*. *Front Cell Infect Microbiol* 9:195. <https://doi.org/10.3389/fcimb.2019.00195>.

The final bacterial concentration of each experimental group is not given. Also, conjugation experiment in liquid culture might be more meaningful.

Response : Thank you for your valuable suggestion. The density in the article is the final bacterial density. The conjugation experiment in liquid culture is the goal of our subsequent experiment.

temperature change :

Need to indicate the temperature fluctuation range in the actual experiment.

Response: Thank you for your suggestion. We have revised it in

"Materials and Methods" (Lines 375-376).

Since the final bacterial concentration at the end of the experiment is not given, please explain how to rule out the possibility that different culture temperatures lead to different bacterial densities, which may affect the conjugation efficiency.

Response: In the experiment of the influence of temperature on conjugation transfer efficiency, the initial bacterial concentration was 10^8 CFU/mL, and the final bacterial concentration was 10^{11} CFU/mL. This bacterial concentration was in a sticky state, and the rate of bacterial proliferation was very slow, which had little influence on conjugation transfer efficiency.

nutrient level

Lack of quantifiable indicators, such as calorie, protein, carbohydrate, fat, etc (shrimp feed filter, the composition of feed shall be indicated).

Response :Actually, we just want evaluate the risk of conjugative transfer in different survival factors. Sterile seawater and 2216E broth are used to simulate seawater condition. 1/10 diluted 2216E broth is used to simulate desalination condition. Shrimp feed filtrate is used to simulate the aquaculture condition after feeding. The filtrate of shrimp hepatopancreas is used to simulate the condition inside the host. LB and M9 broth are used to simulate the lab condition. As for the quantifiable indicators, it can be further studied in the future work.

Another question: the author specially emphasized 2% NaCl in this section. In the dilution experiment, are the salinities of various media the same? Is osmotic pressure the same?

Response : Actually, we just want evaluate the risk of conjugative transfer in different survival factors. Sterile seawater and 2216E broth are used to simulate seawater condition. 1/10 diluted 2216E broth is used to simulate desalination condition. Shrimp feed filtrate is used to simulate the aquaculture condition after feeding. The filtrate of shrimp hepatopancreas is used to simulate the condition inside the host. LB and M9 broth are used to simulate the lab condition. We can design new experiment to compare the impact of the salinity and osmotic pressure on conjugation.

Line 237 why use LG+G+I+F model? needs explanation.

Response : Before construct the phylogenetic tree, the amino acid substitution model should be evaluated and LG+G+I+F model is the best model for this tree. MEGA now contains methods for selecting the best-fit substitution model(s), estimating evolutionary distances and divergence times, reconstructing phylogenies, predicting ancestral sequences, testing for selection, and diagnosing disease mutations.

Reference: Caspermeyer J. (2018). MEGA software celebrates silver anniversary. *Mol Biol Evol.* 35(6):1558–1560.

Tamura, K., Stecher, G., and Kumar, S. (2021). MEGA11: Molecular

Evolutionary Genetics Analysis Version 11. Mol Biol Evol 38(7),
3022-3027. doi: 10.1093/molbev/msab120.

Line 307 "Inoculated broth was cultured in a shaker for 24 h."
temperature is need to specified.

Response: Thanks for your reminding and we have revised to “The
inoculated broth was cultured in a shaker for 24 h at 28°C” (Line 343).

Table 1: only identity value , coverage and E-value are needed.

Response: Thanks for your suggestion and we have revised it in Table 1.

Resolution for Fig.2 is low. To exhibit each branch, a traditional tree is
needed instead of circle tree.

Response: We have redrawn Figure 2 and resubmitted.

September 1, 2022

Dr. Xuan Dong
Yellow Sea Fisheries Research Institute, Chinese Academy of Fishery Sciences
No. 106, Nanjing Road
Qingdao
China

Re: Spectrum01702-22R1 (Conjugative transfer of AHPND-causing pVA1-type plasmid is mediated by a novel self-encoded type IV secretion system)

Dear Dr. Xuan Dong:

Your manuscript has been accepted, and I am forwarding it to the ASM Journals Department for publication. You will be notified when your proofs are ready to be viewed.

Sincerely,

Jennifer Auchtung
Editor, Microbiology Spectrum

Journals Department
Supplemental Material FOR Publication: Accept